# DispHScan: A Multi-Sequence Web Tool for Predicting Protein Disorder as a Function of pH

**DOI:** 10.3390/biom11111596

**Published:** 2021-10-28

**Authors:** Carlos Pintado-Grima, Valentín Iglesias, Jaime Santos, Vladimir N. Uversky, Salvador Ventura

**Affiliations:** 1Institut de Biotecnologia i Biomedicina, Departament de Bioquímica i Biologia Molecular, Universitat Autònoma de Barcelona, Bellaterra, 08193 Barcelona, Spain; Carlos.Pintado@uab.cat (C.P.-G.); valentin.iglesias.mas@gmail.com (V.I.); Jaime.Santos@uab.cat (J.S.); 2Department of Molecular Medicine, USF Health Byrd Alzheimer’s Research Institute, Morsani College of Medicine, University of South Florida, Tampa, FL 33612, USA; vuversky@usf.edu

**Keywords:** conditional disorder, pH, sequence analysis, protein structure, bioinformatics

## Abstract

Proteins are exposed to fluctuating environmental conditions in their cellular context and during their biotechnological production. Disordered regions are susceptible to these fluctuations and may experience solvent-dependent conformational switches that affect their local dynamism and activity. In a recent study, we modeled the influence of pH in the conformational state of IDPs by exploiting a charge–hydrophobicity diagram that considered the effect of solution pH on both variables. However, it was not possible to predict context-dependent transitions for multiple sequences, precluding proteome-wide analysis or the screening of collections of mutants. In this article, we present DispHScan, the first computational tool dedicated to predicting pH-induced disorder–order transitions in large protein datasets. The DispHScan web server allows the users to run pH-dependent disorder predictions of multiple sequences and identify context-dependent conformational transitions. It might provide new insights on the role of pH-modulated conditional disorder in the physiology and pathology of different organisms. The DispHScan web server is freely available for academic users, it is platform-independent and does not require previous registration.

## 1. Introduction

Intrinsically disordered proteins (IDPs) do not have a static tertiary structure but fluctuate between unfolded and partially folded states. This conformational plasticity is fundamental for their biological activity as regulatory elements in the cell [1]. In addition, disordered tags are commonly used in the biotechnological industry to expose functional epitopes, increase protein solubility, or engineer pharmacological properties [2].

The unfolded nature of IDPs is encoded in their primary structure [3] and, therefore, can be predicted from the polypeptide sequence. Multiple computational tools, exploiting diverse physicochemical protein principles, have succeeded in identifying disordered proteins in standard conditions [4]. IDPs amino acid side chains are largely exposed to the solvent, which makes these proteins intrinsically sensitive to environmental fluctuations. Thus, their disordered/ordered state is often modulated by factors extrinsic to the sequence, such as binding partners, counterions, pH, or macromolecular crowding [5], resulting in conditional disorder. The biological relevance of an increasing number of conditionally disordered proteins [6] aims for the development of context-dependent prediction methods to forecast such conformational transitions.

In a recent study, we modeled protein disorder as a function of pH and developed a disorder predictor able to anticipate disorder-to-order transitions in IDPs named DispHred [7]. The DispHred algorithm was based on the realisation of Uversky and his coworkers that protein disorder could be anticipated by analyzing two simple biophysical properties: protein net charge and hydrophobicity [3]. IDPs populate a defined region in the charge–hydrophobicity space (C-H), so it is possible to discriminate disordered and folded proteins by applying a simple boundary condition in the C-H diagram. This method has been implemented in several computational predictors [8], yet it was only validated under physiological conditions at neutral pH. With DispHred, we were able to reformulate the C-H diagram, recalculating both charge and hydrophobicity as a function of pH, which allowed single protein disorder predictions at different pHs.

Despite the novelty of the method, our algorithm was still limited to the analysis of a single unique polypeptide sequence at a time, precluding its application to study pH-dependent protein transitions in either large or biologically and biotechnologically relevant datasets.

Here, we address this need by implementing the DispHScan web server, a first computational tool to predict protein disorder and folding transitions as a function of pH for multiple sequences. The DispHScan server does not require registration and is free for academic users, providing an easy and fast means to explore pH-dependent conditional disorder in proteins.

## 2. Methods

### 2.1. DispHScan Pipeline

DispHScan outlines pH-dependent disorder in a defined pH interval and identifies conformational transitions for multiple sequences. The server computes the charge and hydrophobicity as a function of pH and applies a linear boundary condition to discriminate the folding state of each individual sequence (Figure 1). The details of the DispHScan pipeline architecture are described below:

*Input interface*: Users can either upload a file or paste the sequences in FASTA format on the specified area. Afterward, the pH interval—with the selected step and window size—must be defined (default values are 0.5 and 51, respectively). Alternatively, the option of predicting disorder at only 1 pH is also available for selection. For clarity, five sequences with defined pH ranges will be uploaded as model IDPs upon clicking the *Example* button.

*Hydrophobicity*: Sequence lipophilicity is calculated using a pH-dependent scale of amino acids based on implicit solvation calculations [9]. A sliding window—whose size is defined by the user—is used to compute local lipophilicities, which are then averaged to obtain mean hydrophobicity. Final scores and global protein analysis remain largely unaffected by the window size; however, it influences local predictions along the sequence and the resulting profile.

*Net Charge Per Residue (NCPR)*: The net charge for each amino acid is calculated with the Henderson–Hasselbalch equation. The global net charge is computed as the sum of all local charges, which is averaged to obtain NCPR.

*Disorder calculation*: Hydrophobicity and NCPR are combined in the linear boundary condition equation described by Santos and their co-workers to discriminate the folding state for each pH in range (DispH score) [7]. The algorithm classifies proteins with positive DispH scores as folded and negatives as unfolded.

*Structural transitions*: Transitions are detected when maximum and minimum DispH scores vary in sign and are outside the margins of the confidence interval (±0.02). Multitransitions are informed when consecutive DispH scores shift their sign if more than one transition has already been recognized. Proteins whose DispH score signs do not vary in the pH interval of study are classified as folded (positive values) or unfolded (negative values).

*Output presentation*: Users can check the results in JSON format or download them in a ZIP file that contains all generated data (CSV and JSON files, tables, and figures). An interactive table is also provided that summarizes the results for each sequence, describing predicted transitions and the corresponding folding state. For each detected transition, the pH at which such transition occurs is specified. Maximum and minimum DispH scores—with their associated pH—are also displayed. Clicking on identifiers will open a figure showing the disorder profile at each pH. If the option of predicting disorder at one pH has been selected, the results summarize the overall DispHScore for each sequence, along with their respective hydrophobicity and NCPR. A representation of the disorder variation per residue at such specific pH is also provided.

### 2.2. Server Implementation

The DispHScan server is built upon the Django 3.0 web framework. The web interface is written in HTML/CSS/JS. In the server side, app scripts are written in Python, using Python 3.7 as the interpreter.

### 2.3. Gene Ontology Annotation

To investigate the nature of transitioning proteins, gene ontology terms were analyzed using the Database for Annotation, Visualization, and Integrated Discovery (DAVID) v6.8 [10]. The complete human proteome (UP000005640) was used to infer enrichments for the different data subsets. Enrichments for GO_direct categories displaying *p*-values < 10^−3^ were considered confident.

## 3. Results

### 3.1. Performance

DispHScan, available online at: http://disphscan.ppmclab.com (accessed on 25 October 2021), is designed to handle large datasets on a wide range of pHs to characterize pH dependence of the conformational state of proteins. Its performance was tested on the human proteome (UP000005640) from pH 0 to pH 14 using a step size of 0.5 units and a window size of 51 residues (Figure 2). The program was able to scan a total of 20,600 sequences, each at 29 different pHs (597,000 data points) in less than 17 h. The output identified 1317 proteins with at least one transition, which accounts for 6.4% of the proteome. Most of them (89.1%) corresponded to a disorder-to-order transition when the pH increased, with 58.6% of the transitions occurring at acidic pH (<6), implying that these proteins would be ordered in a physiologically relevant pH range between 6 and 8. We identified 249 proteins predicted to experiment a conformational transition at close to neutral pH and thus under normal physiological conditions. In addition, 13.2% of the transitioning proteins experiment multitransitions, ranging from 2 to 4 conformational switches (Figure 2).

The analyses revealed significant correlations between the level of disorder and the nature of the transition. On average, proteins with a single transition are more disordered (lower DispH scores) than those that do not present pH-dependent conformational change at neutral pH (Appendix A). Similarly, multi-transition proteins tend to have lower DispH scores than proteins exhibiting a single transition (Appendix A). This suggests an underlying correlation between the propensity to transition and the degree of disorder.

We performed a functional analysis to infer GO annotations on transitioning proteins. As expected for IDPs, we observed that proteins displaying transitions were functionally enriched in intracellular regulatory processes, such as transcriptional regulation or RNA binding (BP_direct and MF_direct terms, *p* = 10^−139^ and 10^−56^ respectively), and they are mainly located in the nucleus and the ribosomes (CC_direct term, 10^−120^ and 10^−48^).

### 3.2. Analysis of Model Organisms

We made use of DispHScan to conduct a pH-dependent disorder analysis in the proteomes of three additional model organisms, including *Escherichia coli* (UP000000558), *Saccharomyces cerevisiae* (UP000002311), and *Caenorhabditis elegans* (UP000001940). The resulting pre-calculated data are available under the *Model data* section of the server. The main outcome, shown in Table 1, reveals a considerable fraction of their proteomes exhibiting pH-dependent disorder, suggesting that it might be an important factor in the dynamic conformation of their IDPs. The proportion of transitioning proteins seems to be lower in *E. coli.* (2.6%) than in the eukaryotic organisms, where they occur in roughly 6% of the proteins, consistent with a lower proportion of IDPs in prokaryotes [11]. The nature of transitions, described in Table 2, includes not only single conformational changes but the potential of multiple transitions occurring in a small fraction of the proteins. Remarkably, the most predominant conformational switch observed is a conditional folding towards more ordered structures occurring in the physiological pH range.

## 4. Discussion

The knowledge of the nature of IDPs we have gained in recent decades has allowed researchers to uncover the crucial role these unstructured polypeptides play in cell functioning [1]. As the field has grown, new computational tools have been developed to anticipate global and local protein disorder propensity from primary protein sequences. They have been instrumental in the study of IDPs, guiding and assisting numerous experimental endeavors. These algorithms allow us to systematically project our knowledge on the disorder properties of defined protein sets into thousands of uncharacterized polypeptide sequences in both time- and cost-effective ways. This capability has been central to understanding the prevalence of disorder at the proteome level [11] and instrumental for incorporating their outputs into more complex predictive pipelines [12,13,14].

DispHred [7] expanded the predictive potential of previous algorithms by considering both the sequence and the specific pH of the solution, allowing the forecasting of pH-dependent order–disorder transitions. Nevertheless, DispHred only permitted single-sequence analysis and was unsuitable for identifying transitioning proteins in large datasets, something that the community soon requested upon its publication. To amend this limitation, we developed DispHScan, a new web tool that exploits the predictive accuracy of our original model [7] but is designed to handle large protein datasets efficiently.

DispHScan provides a free and user-friendly platform for the multi-sequence prediction of pH-dependent disorder. We expect that DispHScan would find application in investigating the role of pH-conditioned disorder in different aspects of cell biology at a proteome-wide level. With this idea in mind, we provide users with precomputed pH-dependent disorder analysis for four model organisms of interest, facilitating rapid access to the respective DispHScan outputs.

DispHScan might also be of interest for biotechnological applications by helping researchers detect stable conformations at pH values far from neutrality, or by assisting in the redesign of protein variants displaying a conformational response to changes in the solution pH. This is especially relevant, considering that protein purification, storage, and formulation often involve pH values away from physiological conditions. Assessing pH-dependent disorder is also valuable for optimizing solvent conditions and to speed up buffer screenings. In addition, this tool can be of help in the study of pH-modulated liquid–liquid phase separation (LLPS), where pH-jumps are used to prevent or trigger LLPS, allowing kinetic analysis in near-native conditions [15].

Overall, we envision that the use of DispHScan might provide new insights into the physiological and pathological role of the solution pH in the conformational plasticity of IDPs and assist scientists and industries in identifying optimal pH conditions for proteins of interest.

## Figures and Tables

**Figure 1 biomolecules-11-01596-f001:**
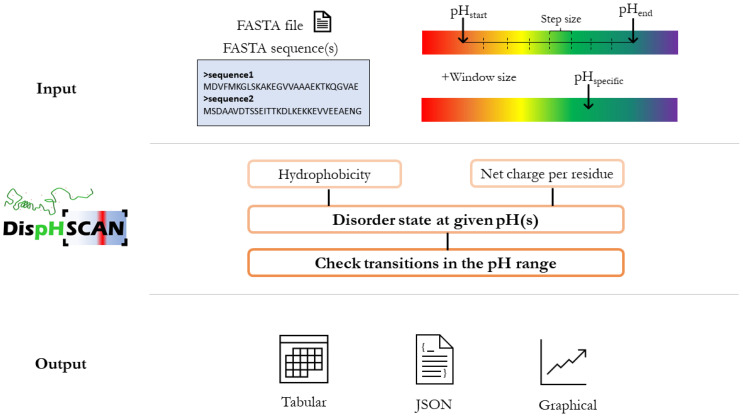
DispHScan pipeline. Users must introduce their sequences in FASTA format and select the pH interval with the desired step and window sizes (default values are 0.5 and 51, respectively). The option of predicting disorder at a single pH is also available. The server computes mean hydrophobicity and NCPR to provide a disorder prediction for each sequence and pH. Possible transitions are checked in the interval of study. The results are represented in both tabular and graphical formats, as well as in JSON; all of them available for download.

**Figure 2 biomolecules-11-01596-f002:**
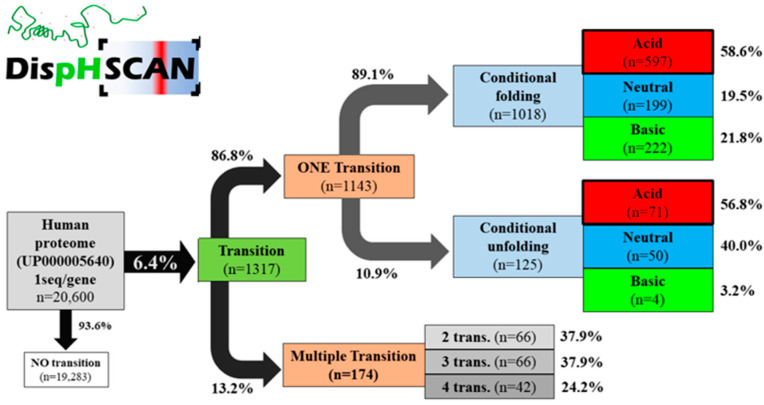
DispHScan statistics on the human proteome. Proteins were classified based on whether they experiment no-, single- or multiple-transitions. Protein switches from disordered states at lower pH to ordered states at higher pH were defined as conditional folding, whereas the reverse transition was defined as conditional unfolding. Depending on the specific pH at which the folding/unfolding transition occurs, they were named acid (<6), neutral (6–8), or basic (>8).

**Table 1 biomolecules-11-01596-t001:** Classification of pH-dependent disorder for the proteomes of four different model organisms. Most proteins remain in a defined conformational state (no transition). However, a smaller but significant proportion of the proteins in each proteome exhibits pH-dependent conditional disorder (transition).

Organism	*n*	No Transition (*n*)	Transition (*n*)	No Transition (%)	Transition (%)
*H. sapiens*	20,600	19,283	1317	93.6	6.4
*E. coli*	5062	4932	130	97.5	2.6
*S. cerevisiae*	6050	5661	389	93.6	6.4
*C. elegans*	19,813	18,735	1078	94.6	5.4

**Table 2 biomolecules-11-01596-t002:** Nature of the pH-dependent transitions in the proteomes of four different model organisms. In most of the cases, proteins undergo a single conformational transition towards a more ordered state (conditional folding). Some proteins might switch their conformation more than once along the pH range (multitransition).

Organism	Single Transition (%)	Multitransition (%)	Conditional Folding (%)	Conditional Unfolding (%)
*H. sapiens*	86.8	13.2	89.1	10.9
*E. coli*	94.6	5.4	88.6	11.4
*S. cerevisiae*	85.6	14.4	82.9	17.1
*C. elegans*	80.1	19.9	86.2	13.8

## Data Availability

Data shown in this study can be accessed under the Model data section of the server’s main page http://disphscan.ppmclab.com/ (accessed on 25 October 2021). Results on the validation of the method can be checked in article [7].

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
