# Peer review of "DispHScan: A Multi-Sequence Web Tool for Predicting Protein Disorder as a Function of pH"

_biomolecules, 2021, doi:10.3390/biom11111596_

Round 1

Reviewer 1 Report

The present work by Pintado-Grima and collaborators builds upon their previous work, mainly the generation of a predictor of pH transitions in IDPs (DispHred). The authors have improved and expanded on this work, creating a computational tool to predict  pH-induced disorder-order transitions in large protein datasets (like proteomes) and including the possibility of scanning a wide pH range. The authors demonstrate the utility of this tool analyzing the proteome of 5 organisms, including humans and model organisms.

This is exactly the kind of tool that protein scientists need right now to supplement the information that can be obtained from the new wave of structure-predicting servers like AlphaFold. The available tools for predicting and understanding features from IDPs have historically lagged when compared to their well-folded relatives, and enabling the possibility of assessing large data sets is exactly the kind of feature that helps pushing the field of IDPs forward. Thus, this is a most welcome utility. I find the prospect of running this sort of analysis on viral proteomes -known for the particularly high percentage of IDPs and IDDs and structure-function versatility in different cellular compartments- particularly interesting.

Minor comments below:

1) Introduction, lines 35-36: The authors should clarify what they mean with "low cooperativity of IDPs' conformational ensembles"

2) Methods, line 70: I suggest writing "default values are 0.5 and 51, respectively"

3) Figure 1: It's hard to distinguish the main features of the "Output" segment. 

4) Figure 2: A large part of the caption repeats information already present in the main text and should be removed.

Reviewer 2 Report

The authors developed a predictor for pH-induced disorder-order transition protein(s), DispHScan. DispHScan is applicable for large-scale predictions such as proteome-wide analysis, and the authors applied DispHScan on the proteomes of several model organisms. Although I am positive for publishing this work on Biomolecules, I would like the authors to consider the following points.

1) The manuscript says that analysis on the human proteome took 17 ours. Do users can conduct same analysis from the web ? If users can submit a large number of sequences, how to obtain the result ? I could not find a box for e-mail on the submission page, and do users need to keep the web page open to obtain the results ?

2) Related to the point 1), is there any limitation of the number of sequences ?

3) DispHScan uses the default window size of 51. When we submit a protein shorter than 51 residues, does DispHScan predict it well ?

4) Related to the point 3), is there any guideline to select window size ? For instance, when we take shorter window, the result tends to be something... etc.

5) The results on the proteome analysis are interesting. I want to know additional information. For example, what kind of proteins showing one transition and multiple transition. This kind of information can be known by referring UniProt (e.g. GO term etc.), and the IDR percentage of these proteins by applying some IDR prediction methods.

Reviewer 3 Report

The manuscript submitted by S. Ventura and colleagues describes a server designed for predicting pH-dependent order-disorder transitions in large sets of protein sequences. It was used to analyze four proteomes (H. sapiens, S. Cerevisiae, C. elegans, and E. coli).

The algorithm behind these computations was published by S. Ventura about one year ago (reference 7).

The manuscript is well written and the results are interesting and useful.

I have only a couple of observations.

a) The first sentence of the Discussion (The increasing knowledge on the nature of IDPs we have gained in the last decades 185 has allowed researchers to uncover the crucial role these unstructured polypeptides play 186 in cell functioning [1].) mentions a rather old reference (Dunker, A.K.; Brown, C.J.; Lawson, J.D.; Iakoucheva, L.M.; Obradovic, Z. Intrinsic disorder and protein function. 229 Biochemistry 2002, 41, 6573-6582, doi:10.1021/bi012159+). Might it be possible to use a more recent one? This is necessary for younger readers.

b) Again in the Discussion (lines 203-205), the Authors write that “We expect that DispHScan would find application in investigating the role of pH-conditioned disorder in different aspects of cell biology at a proteome-wide level.” Well, I would go further. This might help in finding proteins biotechnologically interesting, stable at unusual pH values. In addition, this might help structural biologists to select the optimal pH values where to examine their samples.

Round 2

Reviewer 2 Report

The authors revised based on the comments of the first round, and the revision can be acceptable. Then, I believe the manuscript can be accepted to be published  in Biomolecules.